# Bioactive Compounds of Porcine Hearts and Aortas May Improve Cardiovascular Disorders in Humans

**DOI:** 10.3390/ijerph18147330

**Published:** 2021-07-08

**Authors:** Irina Chernukha, Elena Kotenkova, Svetlana Derbeneva, Daniil Khvostov

**Affiliations:** 1V. M. Gorbatov Federal Research Center for Food Systems of RAS, Experimental Clinic and Research Laboratory for Bioactive Substances of Animal Origin, Talalikhina St. 26, Moscow 109316, Russia; imcher@inbox.ru; 2Federal Research Center of Nutrition, Biotechnology and Food Safety, Department of Cardiovascular Pathology, Kashirskoe Sh. 21, Moscow 115446, Russia; sderbeneva@yandex.ru; 3V. M. Gorbatov Federal Research Centre for Food Systems of RAS, Laboratory of Molecular Biology and Bioinformatics, Talalikhina St. 26, Moscow 109316, Russia; d.hvostov@fncps.ru

**Keywords:** hyperlipidaemia, hypertension, cholesterol, bioactive peptides, trial

## Abstract

Functional foods promote health benefits in human metabolism, with bioactive compounds acting as therapeutic agents. The aim was to investigate the biological effects of a pâté made of pork hearts and aortas, minced, sterilised and packed in tins. Adults (61–66 years old) with a body mass index of 26.4–60.7 kg/m^2^ (n = 36) were randomly divided into two groups: one group consumed a low-calorie diet (LCD), while the other consumed an LCD with the developed meat product (MP) for 28–30 days. Serum biochemical parameters, anthropometry and blood pressure were measured. Consumption of an LCD + MP by experimental group participants helped to maintain reduced cholesterol levels. The difference in total cholesterol was significantly different (*p* = 0.018) from that of the control group, mainly due to the difference in low-density lipoprotein cholesterol (*p* = 0.005). Six peptides with potential cholesterol-binding properties and four peptides with potential antioxidant activity were identified in the MP, while elevation of the content of two peptides with potential angiotensin-converting enzyme-inhibitory activity was detected in patients’ plasma. Intervention with the MP can be considered as a supportive therapy to the main treatment for medical cardiovascular diseases due to a positive effect on serum cholesterol.

## 1. Introduction

Foods have been used in disease prevention for many years. Currently, there is a trend to discover the curative properties of everyday foods. Functional foods (FF) can be defined as those providing health benefits beyond basic nutrition and include whole, fortified, enriched or enhanced foods that have a potentially beneficial effect on health when consumed as part of a varied diet on a regular basis at an effective level, focusing on the prevention of chronic, non-infectious, nutrition-related diseases [1,2]. Despite the fact that the majority consider FF to have both preventive and curative properties [3,4,5,6,7], some researchers state that FF do not cure or prevent illnesses by themselves and are not essential to the diet [8].

Meat is one of the most consumed product types in both Western and Eastern diets. Consumers’ attitudes towards meat intake strongly depend on food behaviour as well as understanding of the functionality and potential health-promoting ability of meat products [9,10]. Meat-based FF provide the consumer/patient with the necessary nutritive biologically active components that are capable of improving metabolic disorders and providing health benefits in human metabolism, with bioactive compounds acting as therapeutic agents [11]. Meat is a good matrix for FF development, and many technological approaches have been investigated, such as composition modification, fermentation and the introduction of functional additives [3,12,13]. Moreover, meat is a good source of bioactive peptides present in animal tissue natively or latently and released during processing (heat or enzymatic treatment) or digestion [14,15,16,17]. These peptides have the potential to positively affect metabolic disorders, such as obesity, cardiovascular diseases (CVD) and neurodegenerative diseases, by lowering the comorbidity burden [18,19,20].

While meat consumption is increasing, especially in developing countries, offal (animal by-product) consumption has declined, despite its low prices [21]. The term animal by-product denotes a product of animal origin that is not intended for human consumption. Animal tissues such as muscles, bones, blood, brain and glands are well-known sources of biologically active compounds [15,22,23,24], but aortas have been poorly investigated in this context.

Earlier, we confirmed the health-promoting properties of raw porcine and cattle hearts and aortas as well as products made from porcine hearts and aortas in a rat model of hyperlipidaemia [25,26], and demonstrated that these tissues are a source of bioactive components with hypolipidemic characteristics [27]. However, despite very promising results obtained using animal models, the data were insufficient to predict the effects in a diseased human population.

We hypothesised that bioactive peptides activated in porcine heart and aorta tissues during processing could mitigate metabolic syndrome in people with obesity. The aims of the present study were to examine the nutritional content of a specialised pâté, to analyse its effect on patients with a high body mass index and to assess changes in anthropometric parameters, blood pressure, serum cholesterol profiles and liver status indices, as well as to obtain relevant evidence of the curative properties of the meat product developed through discovery-based research.

## 2. Materials and Methods

### 2.1. Meat Product

The meat product (MP), developed for specialised nutrition, was produced at ZAO Yoshkar-Olinskiy Myasokombinat. Porcine hearts were chopped into a particle size of 2–3 mm and salted for 12 h. Porcine aortas were chopped into a particle size of 2–3 mm and then homogenised in a cutter at 3000 rpm for 2–3 min. Minced hearts with the juice were quantitatively transferred to the cutter and homogenised at 3000 rpm for 6–8 min (aor-ta:heart ratio 1:3). The mince was packed in tins and sterilised at 115 °C and a pressure of 0.23 MPa for 40 min. The meat product contained 17.53 ± 0.95% protein, 3.82 ± 0.13% fat, 0.305 ± 0.015% sodium chloride and 2.35 ± 0.25% starch.

### 2.2. Study Design

A randomised controlled trial (RCT) study design was used. Recruitment took place from 1 June 2019 to 15 October 2019 at the Department of Cardiovascular Pathology of the Federal Research Centre of Nutrition and Biotechnology. Patients were randomised into two groups by flipping a coin. The participants were not blinded to their group assignment. After hospitalisation, the patients signed the informed consent form, and then they were assigned a sequential number. The work was carried out in accordance with the protocol approved at the profile commission meeting of the Health Care Expert Council of the Ministry of Health of Russia, Protocol No. 5 of 14 June 2018. The Ethics Commission acts in accordance with the RF rules and Declaration of Helsinki Ethical Principles for Medical Research Involving Human Subjects, 2013. The study protocol was also approved by the Ethics Committee of the Federal Research Centre of Nutrition and Biotechnology (Protocol No. 7 of 3 December 2018). During the study, 14 participants were excluded from the study for various reasons (Figure 1).

Participants in the control group (n = 18) consumed a standard hyposodium low-calorie diet (LCD) for 28–30 days (7–10 days as an inpatient and 18–20 days as an outpatient), while subjects in the experimental group (n = 18) received the LCD and 100 g MP per day. The LCD is a diet with a significant restriction of fat and easily digestible carbohydrates that contains a normal level of protein and complex carbohydrates, with an increased amount of dietary fibre and a reduction in the amount of table salt (3–5 g/day). The chemical composition of the LCD and the modified diet with the MP is presented in Table 1.

Meats were boiled, stewed, baked and steamed before consumption. The food temperature ranged from 15 to 60–65 °C. The number of meals was 4–6 per day. Each participant was counselled to consume a version of the LCD during the outpatient period; participants belonging to the experimental group were provided MP. Monitoring of compliance with diet therapy was carried out only at the subsequent visit by personal interview. If it turned out that the participant did not comply with recommendations during the outpatient period, they were excluded from the analysis.

Blood samples for biochemical studies were taken on days 0 (T0), 7–10 (T1) and 28–30 (T2) on an empty stomach and in the morning. Body weight and blood pressure were also measured. The treatment groups did not differ according to baseline characteristics.

### 2.3. Participants

Recruitment took place from 1 June 2019 to 15 October 2019 at the Department of Cardiovascular Pathology of the Federal Research Centre of Nutrition and Biotechnology. The 50 included participants were men and women (61–66 years old) with a body mass index (BMI) of 26.4–60.7 kg/m^2^. These patients had been hospitalised for a course of specialised diet therapy. They were hospitalised as volunteers based on their desire to change food behaviour and improve state of health. The inclusion criteria were atherosclerosis of the heart and blood vessels (ischemic heart disease and/or atherosclerosis of the brachiocephalic arteries and/or atherosclerosis of the lower limb arteries). Ischemic heart disease was diagnosed based on the presence of a combination of clinical symptoms, anamnesis data, the results of exercise tests (bicycle ergometry or treadmill test), as well as coronary angiography data. Atherosclerosis of the brachiocephalic arteries and arteries of the lower limbs was diagnosed by ultrasound examination or anamnestic data.

The exclusion criteria were diabetes mellitus, pregnancy and/or breastfeeding, anaemia, fever, exacerbation of a chronic disease, chronic renal failure, acute and/or decompensated chronic cardiovascular disease, medications (treatment with any other test drug within the last 30 days prior to inclusion in this study, drug therapy with drugs containing sibutramine and/or orlistat) and the use of dietary supplements to control or reduce body weight.

Information on the ethnicity/race of patients was not obtained. All participants belonged to the middle class and had low physical activity level. 

### 2.4. Anthropometric and Blood Pressure Measurements

Height was measured with a standard stadiometer, and body weight was measured using an Inbody 770 bioimpedance analyser (InBody Co. Ltd., Seoul, South Korea). Blood pressure (BP), systolic blood pressure (SBP) and diastolic blood pressure (DBP) were measured using a Little Doctor blood pressure cuff (Little Doctor International (S) Pte. Ltd., Singapore). The body mass index (BMI) was calculated using the Quetelet formula according to the following equation:BMI = body weight (kg)/height (m^2^),(1)

### 2.5. Blood Sampling 

Blood samples for biochemical studies were taken on day 0 (baseline, T0), 7–10 (T1) and 28–30 (T2). Venous blood samples were collected (after a 12 h fast) in a 4 mL vacuum tube with a coagulant (Improvacuter, Guangzhou Improve Medical Instruments Co., Ltd., Guangzhou, China) for biochemical parameter determination and in 4 mL vacuum tube with K2EDTA (Improvacuter, Guangzhou Improve Medical Instruments Co., Ltd., China) for bioactive peptide identification. The blood samples were centrifuged (1500× *g*, 10 min, 4 °C) and the serum/plasma was transferred to microcentrifuge tubes and stored at −30/−80 °C until further analysis. 

### 2.6. Biochemical Analysis

Biochemical investigations were carried out on a BioChem FC-360 automatic analyser (HTI, North Attleborough, MA, USA) according to the instructions supplied with the measurement kits (HTI, North Attleborough, MA, USA). Total cholesterol (TCL), low-density lipoprotein cholesterol (CL LDL), high-density lipoprotein cholesterol (CL HDL), glucose, creatinine and urea (BUN) levels were measured in serum. The activity of lactate dehydrogenase (LDH), creatine phosphokinase (CPK), alkaline phosphatase (AlkPhos), gamma-glutamyl transferase (GGTP), aspartate aminotransferase (AST) and alanine aminotransferase (ALT) were also measured in serum. The atherogenic index (AI) and the de Ritis ratio were calculated according to the following equations:AI = (TCL–CL HDL)/CL HDL,(2)
de Ritis Ratio = AST/ALT,(3)

### 2.7. Fatty Acid Analysis and Cholesterol Determination in the Meat Product

Lipid isolation from the developed product was carried out by chloroform/methanol extraction according to the Folch method. The purity of the isolated lipids was tested using thin-layer chromatography. Determination of the fatty acid composition was carried out on an HP 6890 gas chromatograph (Hewlett Packard, Palo Alto, CA, USA), and fatty acids were identified by comparison with the software database. Cholesterol determination was performed on an Agilent 7890A gas chromatograph (Agilent Technologies, Santa Clara, CA, USA) with an Agilent 5975C mass spectrometric detector (Agilent Technologies, USA), an HP5MS capillary column (Agilent Technologies, USA) measuring 30 mm × 0.25 mm × 0.25 µm and helium as the carrier gas. Determination of the fatty acid composition and cholesterol content was performed according to the reported method in the literature [28].

Classic indices such as ΣSFA, ΣMUFA, ΣPUFA, Σn–6 PUFA and Σn–3 PUFA were calculated, and the PUFA/SFA, index of atherogenicity (IA), the hypocholesterolemic/hypercholesterolemic ratio (HH) and the unsaturation index (UI) were calculated according to the following equations [29]:n–6 PUFA/n–3 PUFA = ΣPUFA/ΣSFA,(4)
IA = [C12:0 + (4xC14:0) + C16:0]/ΣUFA,(5)
HH = (cis-C18:1 + ΣPUFA)/(C12:0 + C14:0 + C16:0),(6)
UI = 1 × (% monoenoics) + 2 × (% dienoics) + 3 × (% trienoics) + 4 × (% tetraenoics) + 5 × (% pentaenoics) + 6 × (% hexaenoics),(7)

### 2.8. Extraction and Sequencing of Biopeptides 

A total of 100 mg of the meat product sample was minced and homogenised with 500 μL of bicarbonate buffer in a MagNA Lyser (Roche, Basel, Switzerland) for 50 s at 6500 rpm. Then, it was centrifuged (15,000× *g* for 10 min at 4 °C). The samples of plasma were warmed to room temperature. Supernatant and plasma were ultrafiltrated in 1.5 mL Amicon tubes (Merk Millipore, Darmstadt, Germany) with a 10 kDa mass filter (15,000× *g* for 10 min at 4 °C) to isolate the peptide fraction. Then, 20% formic acid was added to 300 μL of the sample to stabilise proteins for better ionisation during analysis. Samples of 150 μL were transferred into special vials with inserts for LC-MS/MS analysis.

#### 2.8.1. LC−MS/MS Analysis

For chromatographic analysis, an Agilent 1260 Infinity system (Agilent Technologies, USA) coupled to a time-of-flight mass spectrometric detector Agilent 6545XT AdvanceBio LC/Q-TOF (Agilent Technologies, USA) equipped with DuoJet Stream ESI (Agilent Technologies, USA) and the ion funnel (Agilent Technologies, Waldbronn, Germany. A Poroshell 120 EC-C18 reverse-phase analytical column (2.1 × 100 mm, 1.8 μm) (Agilent Technologies, USA) and a ZORBAX Extend-C18 analytical guard column (4.6 × 12.5 mm, 5 μm) (Agilent Technologies, USA) were used. Elution was carried out in a mobile phase consisting of solvent A, 0.1% *v/v* formic acid (Sigma Aldrich, St. Louis, MO, USA) in deionised water, and solvent B, 0.1% *v/v* formic acid (Sigma Aldrich, USA) in 100% acetonitrile (Sigma Aldrich, USA). Chromatography was carried out in a linear gradient of 3% solvent B over 1 min, from 3% to 60% solvent B over 38.99 min, from 60% to 90% solvent B over 4.99 min, 90% solvent B over 1 min, and then returned to 3% solvent B for 1 min at a flow rate of 400 μL/min. The total analysis time was 50 min. The capillary voltage was 4000 V, the nozzle voltage was 500 V, the drying gas flow was 12 L/min at 275 °C, the gas flow through the casing was 12 L/min at 325 °C, and the atomiser pressure was 35 psi. The high-pressure ion funnel was operated at 175 V high-frequency (RF), the low-pressure funnel at 65 V RF, and the octopole at 750 V. Analyses were performed in full-scan MS mode, as well as in auto MS/MS mode with full scanning from 100 to 1700 *m*/*z*. For the collision cell, an intelligent algorithm for obtaining MS2 ions was chosen based on an increase in the collision energy depending on the mass and charge of the ion. The increased collision energy was: charge +2, slope 3.1, displacement 1; charge ≥ + 3, slope 3.6, offset −4.8. In the MS/MS experiments, nitrogen was used as the collision gas. A reference solution was used—purine ([M + H] + = 121.0509 and Agilent compound HP0921 ([M + H] + = 922.0098)—to calibrate the internal mass throughout the analysis.

#### 2.8.2. Sequencing of Biopeptides

Chromatograms were processed using the Agilent MassHunter BioConfirm software (Agilent B.08.01) according to the peptide search algorithm. All integrated peaks were manually checked to ensure correct peak detection and accurate integration. The files were converted into MGF format for amino acid sequence determination using the Agilent MassHunter Workstation MassProfiler (B.08.01 software, Agilent Technologies, USA) and DeNovoGUI [30]. MS/MS spectra were analysed using the pNovo + (beta) [30] algorithm for Windows. Sequences with a score more than 70.0 were selected. 

The calculations of the peptide relative ratio in the experimental group plasma compared to the control group were carried out in automatic mode using the Agilent MassHunter Workstation MassProfiler (B.08.01 software, Agilent Technologies, USA) by two-way ANOVA, and all samples were previously marked up by groups and time. The algorithms of this software analysed all the mass spectra obtained during the analysis and compared the arithmetic mean intensities of all the obtained ions of peptide molecules between the groups at *p* < 0.05.

Identified peptides were analysed with the databases PepBank [31], BioPep [32] and AHTPDB [33] in accordance with their homologous sequences with known properties. A search for peptide sequences identified in human plasma was carried out in the UniProt DataBase with the species *Sus scrofa* [34]. 

### 2.9. Statistical Analysis

STATISTICA 10.0 software was used for statistical analyses. Results were calculated as the mean ± SD (for meat product analysis) and as the median and 25th–75th percentile (for trial results). Significant differences were tested by non-parametric Mann–Whitney *U* tests for independent variables; Freidman ANOVA (n > 2) and Wilcoxon tests (n = 2) were used for dependent variables. Differences with *p*-values less than 0.05 were considered statistically significant. The Spearman coefficient was used for the evaluation of correlations between parameters within groups.

## 3. Results

The baseline participant characteristics corresponding to T0 (0 days) are presented in all tables with the trial results and did not demonstrate any significant differences.

### 3.1. Anthropometry and Blood Pressure

The control and experimental groups did not differ according to baseline body mass, BMI or systolic and diastolic blood pressure (*p* > 0.05). The body mass of control group patients decreased by 3.1 points (3.0%) at T1 and by 2.5 points (2.4%) at T2, but this reduction was not significantly different (Table 2). 

The body mass decline was more noticeable but also not significant in experimental group patients: the body mass decreases at T1 and T2 were 3.7 points (3.7%) and 3.5 points (3.5%), respectively. The same dynamic was observed concerning BMI changes. The BMI of control and experimental group patients decreased by 1.2 points (3.0%) and 1.4 points (3.4%) at T1 and by 0.9 points (2.3%) and 1.3 points (3.2%) at T2, respectively; this reduction was not significantly different. Despite the more noticeable difference between T2 and T0 in participants who consumed the LCD + MP, there were no significant changes in the experimental and control groups.

Systolic and diastolic blood pressure declined significantly in both groups at T2 (*p* ≤ 0.001) (Table 3). The systolic blood pressure of control group patients at T2 decreased by 20.0 points (13.3%), while the reduction in the experimental group patients was greater and averaged 25.0 points (16.7%). The diastolic blood pressure of control group patients at T2 decreased by 7.5 points (8.3%), while the reduction in the experimental group patients was greater and averaged 10.0 points (12.1%). No significant differences were observed between the control and experimental groups after the outpatient period.

Nevertheless, the differences between timepoints T2 and T0 were more noticeable, but still insignificant, in participants who consumed the LCD + MP and were greater than those in participants who consumed only the LCD, i.e., by 25.0% (*p* = 0.189) and 33.3% (*p* = 0.146) for systolic and diastolic blood pressure, respectively.

### 3.2. Serum Cholesterol Profile

The control and experimental groups did not differ according to baseline TCL, LDL and HDL, or the atherogenic index (*p* > 0.05). The results are presented in Table 4. 

Consumption of the LCD until T1 produced a decrease of 10.9 points (7.8%) in the total cholesterol levels, but these had returned to basal levels by T2 (recovery effect). Cholesterol level changes in the control group were not significant (*p* = 0.179). Statistically significant CL LDL and CL HDL reductions, i.e., by 9.1 points (12.7%) and 2.9 points (6.7%), respectively, were noticed at the T1 timepoint. Both CL LDL and CL HDL returned to their original levels at T2, but the changes in the inpatient period were significant (*p* = 0.029 for both parameters). The serum atherogenic index was reduced by 0.31 points (12.0%) at T1, but this was also not statistically significant (0.066); the atherogenic index increased to basal levels by the T2 timepoint.

LCD + MP consumption resulted in significant changes in serum TCL (*p* = 0.001), cholesterol LDL (*p* = 0.033) and HDL (*p* = 0.017) and the atherogenic index (*p* = 0.003) at the T1 timepoint, which persisted to T2, except for CL HDL, which returned to basal levels by T2. The reduction in total cholesterol levels was 25.0 points (15.8%) at T1, and this increased to 31.1 points (19.7%) at T2, mainly due to the decline in CL LDL by 14.1 points (16.8%) and 25.0 points (7.9%) at T1 and T2, respectively. Noticeable fluctuations in the serum atherogenic index were also observed. This decline amounted to 0.44 points (14.1%) at T1 and increased to 0.62 points (19.8%) at T2.

There were no significant changes in cholesterol levels or the atherogenic index between the control and experimental groups when nominal values were compared. In both groups, the inpatient period (T1) led to positive changes in the serum cholesterol profile, although there were no significant changes in cholesterol levels or the atherogenic index at T1. However, during the outpatient period, TCL, CL LDL and CL HDL, and the atherogenic index in the serum of control group participants returned to basal levels (recovery effect) by T2. In contrast, the consumption of the LCD + MP by experimental group participants helped to maintain the reduction in cholesterol levels, and the difference in TCL was significant (*p* = 0.018) compared with that in the control group, mainly due to the difference in CL LDL (*p* = 0.005).

### 3.3. Serum Biochemical Parameters

The control and experimental groups did not differ regarding baseline glucose, creatinine and urea (BUN) levels, or the activity of LDH, CPK, AlkPhos, GGTP, AST and ALT and the de Ritis Ratio. The results are presented in Table 5.

Consumption of the LCD until T1 produced a significant (*p* = 0.021) decrease of 4.5 points (4.4%) in the glucose level, which increased by 9.0 points (8.8%) at T2. In contrast, despite a reduction of 10.8 points (9.1%) in glucose levels in participants who consumed the LCD + MP at T1, these later increased above the basal level by 9.0 points (7.6%). There were significant changes in glucose levels in both the inpatient (T1) and outpatient (T2) periods between the control and experimental groups when comparing nominal values. Participants who consumed the LCD + MP demonstrated higher glucose levels, i.e., by 12.6% (*p* = 0.005) at T1 and 29.6% (*p* < 0.001) at T2 timepoints.

There was no significant change in the creatinine levels (*p* = 0.678) of participants in the control group during the trial. Consumption of the LCD + MP led to an elevation in the creatinine levels (*p* = 0.128) in the serum of experimental group participants by 0.033 points (3.9%) at T1, but these returned to basal levels at the T2 timepoint (recovery effect). There was also a significant difference of 14.8% (*p* = 0.048) in creatinine levels between the control and experimental groups while comparing nominal values during the inpatient period (T1), although no significant difference was observed in the outpatient period (T2). The same tendency was revealed concerning the urea concentration. Urea levels of control group patients at T1 decreased by 3.8 points (9.7%), but it returned to basal levels at T2. In contrast, the consumption of the LCD + MP led to an elevation in urea levels (*p* = 0.042) in the serum of experimental group participants by 10.9 points (23.6%) at T1, but they returned to basal levels at T2 (recovery effect).

There was also a significant difference of 37.0% (*p* = 0.008) in urea levels between the control and experimental groups when nominal values were compared in the inpatient period (T1), although no significant differences were observed in the outpatient period (T2).

Consumption of the LCD led to an insignificant (*p* = 0.358) decrease of 7.2 points (5.3%) in LDH activity at T1, which increased to basal levels at T2. The reduction in LDH activity in the serum of participants who consumed the LCD + MP during the inpatient period (T1) was greater and averaged 11.8 points (9.0%), while after the outpatient period (T2) it exceeded baseline by 11.9 points (9.0%), which made the changes between periods significant (*p* = 0.025).

There were no significant changes in CPK activity (*p* = 0.801) in control group participants during the trial. Consumption of the LCD + MP did not lead to noticeable changes in CPK activity in the serum of participants at T1 and T2. There were no significant changes in LDH and CPK activities between the control and experimental groups when nominal values were compared at T1 and T2.

There were no significant changes in alkaline phosphatase activity (*p* = 0.678) in participants of the control group during the trial. Consumption of the LCD + MP at T1 produced an insignificant (*p* = 0.066) increase of 4.1 points (5.1%) in alkaline phosphatase activity, which was maintained at T2 and averaged 4.7 points (5.8%). Nevertheless, there were no significant changes in LDH and CPK activities between the control and experimental groups when nominal values were compared at T1 and T2, but there were differences after the inpatient period (*p* = 0.031). Consumption of the LCD and the LCD + MP led to a reduction in GGTP activity in serum in both the control and experimental participants by 1.8 points (8.3%) and 2.0 points (9.7%), respectively, at T1, but these levels were slightly elevated above basal levels at T2. Changes within groups and between the control and experimental groups were not significant.

No marked changes were observed in AST and ALT activities in either the control or experimental groups during the trial. Nevertheless, a minor elevation was noted in ALT activity, by 2.2 points (1.5%), in the serum of participants who consumed the LCD at T2, which led to a reduction in the de Ritis ratio by 0.26 points (21.0%), but this was not significant. Consumption of the LCD + MP during the trial promoted a homogeneous AST and ALT balance, demonstrated as a reduction in the de Ritis ratio by 0.22 points (16.4%) at T2, while nominal values at T1 and T2 were the same (1.42 and 1.43); the difference at T1 was on average −0.09, and corresponded to an elevation in the ratio of 6.7% within the group. Nevertheless, there were no significant changes in AST and ALT activities or in the de Ritis ratio between the control and experimental groups when both nominal values and differences at T1 and T2 were compared.

In our study, the Pearson correlation of differences in anthropometric changes between T0 and T2 was not stronger in the control group compared with experimental group participants (Table 6). Nevertheless, there was a correlation observed between HDL level changes and the body mass/BMI in the experimental group.

It was also shown that the reduction in the serum atherogenic index, according to the Pearson coefficients, was mainly correlated with changes in total cholesterol and LDL levels during the inpatient (T1) and outpatient periods (T2) for both the control and experimental groups (Table 7).

The Pearson correlations between differences in serum cholesterol profile and age or gender were not significant during the inpatient (T1) and outpatient periods (T2) for both the control and experimental groups (Table 8).

### 3.4. Lipid Profile of Meat Product

The cholesterol content of the developed MP averaged 271 µg/kg. The results of fatty acids analysis are presented in Appendix A: ΣSFA = 40.96 ± 3.01% of total fatty acids, ΣMUFA = 33.38 ± 4.73% of total fatty acids, ΣPUFA = 13.78 ± 2.46% of total fatty acids, Σn–6 PUFA = 12.94 ± 2.60% of total fatty acids, Σn–3 PUFA = 0.25 ± 0.18% of total fatty acids, PUFA/SFA = 0.33 ± 0.04, IA = 0.46 ± 0.04, HH = 2.09 ± 0.23 and UI = 61.95 ± 3.47.

### 3.5. Identification of Bioactive Peptides

The developed MP mainly contained short peptides with molecular weights less than 1000 Da (71%–93%). The MP peptide profile included 12 peptides with molecular weights 1500 Da and higher, 48 between 1000 and 1500 Da, 114 between 500 and 1000 Da, and 64 in the molecular weight range from 100 to 500 Da. Ten peptides (LCDFYNK, LGADYYTK, VPYHLAAAR, LEYFSSQK, LLAYTTKKK, LFDNYNTLK, HNGN, QGEEFCER, WTCTQGPRWK, GLVDQGQHNCACR) were identified in MP. According to the results of a search of the UniProt DataBase, no peptides were a match for the protein sequences of the species *Sus Scrofa*, except HNGN (Appendix A); therefore, the identified peptides are natively present in pig hearts and aortas and are not derivates of known proteins.

In human plasma, using the MassProfiler program, all obtained MS/MS data were used in the analysis. All found biomolecules were analysed, without separation into natural peptides in plasma and peptides from digested food. Three peptides with a calculated relative ratio in the experimental group plasma compared to the control group (*p* < 0.05) were identified by the accuracy of masses not exceeding 3.2 ppm through the PepBank database [31]. Two peptides (KAAAAP and NLHLP) with angiotensin-converting enzyme (ACE)-inhibitory activity were identified, and the amount was elevated significantly up to the T2 timepoint. By contrast, the amount of the identified peptide FVAPW with dipeptidyl peptidase-inhibitory activity was reduced in human plasma of the experimental group up to the T2 timepoint (Appendix A). A search of the UniProt DataBase for *Sus scrofa* peptide sequences identified in human plasma revealed that the peptide sequence KAAAAP is present in 33 proteins, including 10 Dymeclin isoforms, 6 E1A binding protein p400 isoforms, 3 death inducer-obliterator 1 isoforms, 5 MSL complex subunit 1 and PEHE domain-containing protein isoforms, 4 nucleolar and coiled-body phosphoprotein 1 and LisH domain-containing protein isoforms and 2 CCAAT enhancer binding protein alpha and BZIP domain-containing protein isoforms. The peptide sequence NLHLP is present in 57 proteins, including 19 DNA-directed RNA polymerase subunits and its isoforms, 8 proteins (fragments) of the cytochrome P450 family, 5 alpha-1-antitrypsin isoforms, 4 kinases, 4 proteins containing the SH3 domain and demonstrating protein binding activity, 2 GTP-binding proteins, 1 neurobeachin-like protein, and 12 uncharacterised proteins with 90% similarity with the cytochrome P450 or neurobeachin-like protein families. The peptide sequence FVAPW corresponded to only four proteins belonging to metalloproteinase inhibitor 1 isoforms (Appendix A).

## 4. Discussion

The safety of the clinical use of MP as an addition to traditional dietary therapy of patients with atherosclerosis of the heart and blood vessels, manifested by the absence of its negative impact on the indicators of protein metabolism, indicators of the functional activity of the hepatic system, has been proven. Beneficial effects of MP addition to LCD including further improving the hypolipidemic effects of LCD consumption, as well as BP reduction and improvement in anthropometry parameters, were also observed.

A decrease in body weight was not the aim of this trial, because all the parameters of the control and test diets were similar; in fact, the energy value of the LCD + MP was slightly higher, because of the higher daily protein intake by the experimental patients. Total protein intake was 70.0–80.0 g/day for the control and 86.5–98.0 g/day for the experimental group, or when expressed in relation to body mass, it averaged 0.77–0.8 g/kg/day for the control and 0.98–1.02 g/kg/day for the experimental group. The results of one randomised control trial show that an increase in daily protein intake to 1.0–1.2 g/kg/day reduced fat mass and increased the lean mass percentage compared with the group of patients who had a daily protein intake of less than 1 g/kg/day [35]. Remarkably, patients with daily protein consumption over 1.2 g/kg/day did not show better results than the experimental group. Similar results were summarised in a meta-analysis [36], which corresponds to the slight increase in lean body mass in the experimental group in our study (unpublished data). Body mass and BMI are often correlated with cholesterol parameters, mainly with total cholesterol or CL LDL levels [37].

In both groups, a significant decrease in BP was observed, which corresponded to the sodium reduction in both diets [38]. Interestingly, in the experimental group, this reduction was more noticeable, despite the fact that sodium consumption was higher, i.e., 4.0–4.6 g/day for the control and 4.3–4.9 g/day for the experimental group. Presumably, peptides produced during MP processing also contributed to the decline in BP. Moreover, peptides generated from meat proteins are frequently studied [16,19,39,40].

Among the classic nutritional indices, it was observed that the MP was characterised as having a low content of n–3 PUFA similar to that of pork, which is usually considered as a poor source of n–3 fatty acids [41]. The PUFA/SFA and UI contents of the MP were 0.33 ± 0.04 and 61.95 ± 3.47, which are in the lower border of the ranges 0.27–1.26 and 67–124, respectively, corresponding to that of pork [29,42]. The IA of the MP averaged 0.46 ± 0.04 and was not significantly lower than that of pork, which varies between 0.27 and 0.73 with a median of approximately 0.48 and depends on the breed, sex and feed [29,42,43,44]. The HH of the MP was 2.09 ± 0.23, which corresponded to the usual upper limit for pork, which ranges from 1.72 to 2.26 [45,46]. In summary, the developed MP was not characterised as having outstanding anti-atherogenic values of nutritional indices and was approximately similar to pork, with the exception of the cholesterol content, which averaged 271 µg/kg in the MP, significantly lower than that in pork [44].

Our study revealed that consumption of both the LCD and the LCD + MP led to significant changes in the serum cholesterol profile of participants during the inpatient period (T1). However, changes in the experimental group were clearer after the inpatient period (T1) and were maintained after the outpatient period (T2), while the serum cholesterol profile of control participants returned nearly to baseline. Similar results were obtained in other studies on the influence meat products on the serum cholesterol profile during clinical trials, i.e., the consumption of dry-cured ham led to significant reductions in total cholesterol and LDL [47], as did the consumption of a low-fat meat-based product containing walnuts [5].

The correlation between a reduction in the serum atherogenic index and total cholesterol and LDL levels was expectedly revealed. There were no significant correlations between changes in cholesterols parameters and weight, age or gender, which could be explained by absence of difference in baseline characteristics of treatment groups, as well as uniformity within a group. Nevertheless, numerous studies confirmed that sex, age and weight strongly influence the risk of CVD developing [48,49,50]. In our study, the effectiveness of MP as an addition to the LCD was proven for adults of 61–66 years old with a body mass index of 26.4–60.7 kg/m^2^ and pronounced atherosclerosis. Considering the significant influence of sex, age and weight on the development of CVD, the effect of MP on other population groups could differ from our results or be absent entirely, especially concerning severe genetic hyperlipidaemias [51,52]. We did not obtain the information about the ethnicity/race of participants, which could also impact our study results. 

Analysis of biochemical parameters showed that LCD consumption led to a significant reduction in glucose levels, while this parameter increased in the experimental group. The same trend was noted in adults in whom the total and animal protein intakes and the animal protein intake ratio were positively associated with a homeostasis model assessment of insulin resistance [53]. Moreover, a slight increasing in fat and carbohydrates in the diet of the experimental group could also lead to glucose elevation [54]. In addition, patients in the control group had a higher level of glucose at baseline and could be prone to an elevated level of glucose, which could negatively affect the study results. Elevated protein consumption by experimental participants also led to an increase in creatinine and urea after the inpatient period (T1), which returned to baseline after the outpatient period (T2). However, a systematic review of renal health in healthy individuals associated with protein intake above the US recommended daily allowance in randomised controlled trials showed that increased protein intake ≥10% higher than 0.8 g/kg/day had little or no effect on blood markers of kidney function [55].

There were no significant changes in LDH and CPK activities between the control and experimental groups when nominal values were compared at T1 and T2, nor in transaminase activities. No changes were observed in enzyme activities, which confirmed the absence of toxic effects of the developed meat product on cardiac and liver tissues. The results of one study confirmed that elevated transaminases are associated with a high-protein diet [56], but in our study, protein intake was not as high, and there were no observed changes in liver parameters. 

Most studies in the field of functional meat product processing include technological approaches such as composition modification, fermentation and the introduction of functional additives [3,12,13]. Nowadays, the trend of using animal offal in food production is increasing in popularity, but this practice is still not widespread [22,23,24,57] and depends on traditions and religion [58], as well as legacy regulations [59]. Moreover, the tissue of certain internal organs is characterised by a unique proteome and peptidome, which are involved in the maintenance of normal physiological conditions [60]; this makes it more interesting to evaluate the generation and accumulation of specific peptides during processing.

The highly conserved peptides among those that interact with cholesterol have the general sequence L/V-–X_1–5_–Y–X_1–5_–R/K–, where X_1–5_ can be any sequences of amino acids from one to five residues [61]. It has been suggested that the amino acid residues QG may be a contributing factor to the uptake of radicals; therefore, amino acid sequences containing the QG fragment can exhibit antioxidant properties [62]. There were six identified peptides with potential cholesterol-interacting properties (LCDFYNK, LGADYYTK, VPYHLAAAR, LEYFSSQK, LLAYTTKKK, LFDNYNTLK) and four with potential antioxidant ones (HNGN, QGEEFCER, WTCTQGPRWK, GLVDQGQHNCACR) in MP. In human plasma of the experimental group, two peptides (KAAAAP and NLHLP) with ACE-inhibitory activity were identified, and the amount was elevated significantly up to the T2 timepoint. KAAAAP was previously identified in dry-cured ham (Landrace and Large White crossbreds and paternal line purebred Duroc) and demonstrated an IC_50_ value of 19.79 µM [63]. A peptide search for KAAAAP in *Sus Scrofa* proteins resulted in 33 proteins with the same sequence, with Dymeclin family predominance, some isoforms of which have high expression level in the right coronary artery, left ventricle free wall, heart left ventricle, coronary artery, heart ventricle and cardiac muscle (myocardium). Such proteins as E1A binding proteins p400, PEHE domain-containing proteins, death inducer-obliterator 1, MSL complex subunit 1, nucleolar and coiled-body phosphoprotein 1 and BZIP domain-containing protein are also expressed in the heart and endocardial endothelium, but with lower intensity (Appendix A). Interestingly, the peptide NLHLP with ACE-inhibiting bioactivity elevated in the plasma of experimental group patients was mainly identified in dairy products [64,65]. A peptide search for NLHLP in *Sus Scrofa* proteins resulted in 57 proteins with the same sequence, with a predominance of DNA-directed RNA polymerase subunits and its isoforms, some isoforms of which have medium or low expression in the heart, endocardial endothelium and heart left ventricle. Other proteins are characterised mainly by antioxidant or protein-binding activity and are characterised as having medium, low or no expression in the *Sus scrofa* heart and aorta (Appendix A). FVAPW is known as an inhibitor of the DPP-III enzyme [66]. By contrast, the FVAPW content was reduced in the plasma of experimental group patients. A peptide search for FVAPW in *Sus Scrofa* proteins resulted in four proteins belonging to metalloproteinase inhibitor 1 isoforms, among which only one has a medium expression level in the heart left ventricle (Appendix A). Bioactive peptides usually have a short half-life [67,68], and therefore it is obvious that the ten identified peptides in MP were not detected in patients’ plasma. Meanwhile, according to Appendix A, peptides identified and elevated in plasma of patients belonged to experimental group are encoded in the sequence of proteins expressed in the heart and endocardial endothelium and could release during digestion. 

The effect of the MP developed during this RCT was mainly targeted to the serum cholesterol profile, which allowed us to hypothesise that the six identified peptides mainly demonstrate cholesterol-lowering activity, which is stable over time. Moreover, two peptides with ACE-inhibitory activity were elevated in human plasma of the experimental group, which may be potential protein precursors in *Sus scrofa* heart and aorta tissues or may have arisen due to the influence of other bioactive components on the human body. 

## 5. Conclusions

Very few patents describing the extreme interest in the study of animal by-products date back to the 1960–1970s. This could be true for Western countries, but not for Russia, where the use of non-edible animal products is traditionally of interest; the use of offal bioactive compounds as a curative and/or functional food ingredient has been a topic of research for over 30 years. These results will improve our understanding of the relationship between diet and health. In this study, we used neither animal production nor technological strategies as an instrument to increase the quantity of biologically active compounds in the finished product. We have shown that animal by-products/offal can contain compounds in sufficient amounts that may have a pronounced health-promoting effect. Interventions including the developed product may be considered as supportive therapies in medical treatment mainly directed at lowering serum cholesterol.

## Figures and Tables

**Figure 1 ijerph-18-07330-f001:**
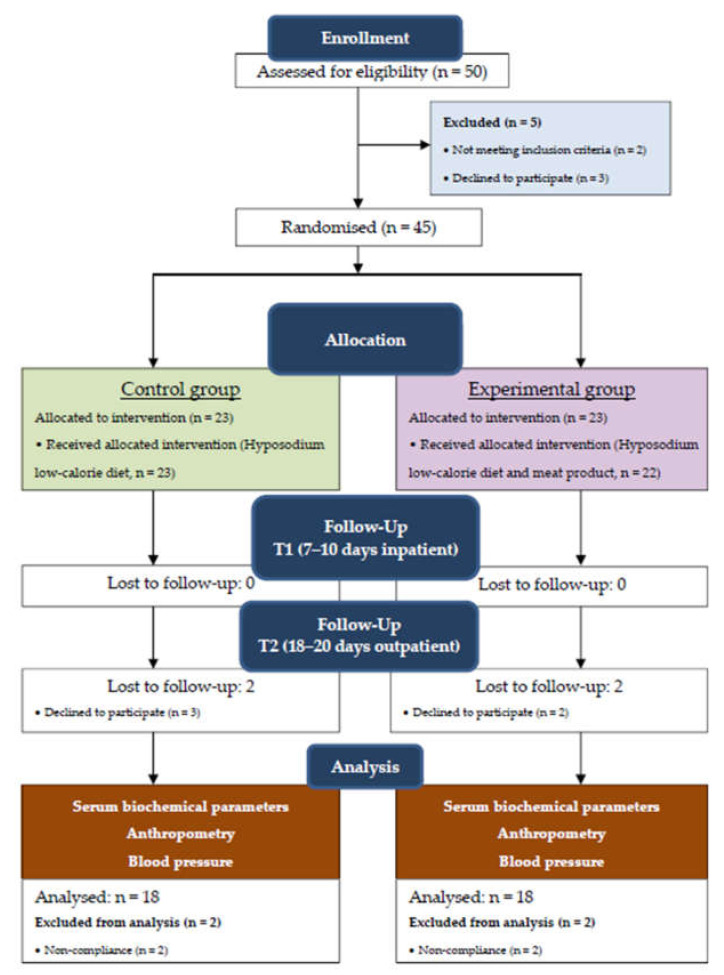
CONSORT flow diagram of participants’ progress through each stage of the study.

**Table 1 ijerph-18-07330-t001:** Chemical composition of control and experimental diets during the trial.

Chemical Composition	LCD	LCD + MP
Energy value, kcal/day	1350.0–1550.0	1437.0–1682.0
Proteins, g/day	70.0–80.0	86.5–98.0
Fat, g/day	60.0–70.0	63.0–76.0
Saturated fatty acid, g/day	20.6–24.0	21.8–25.5
Polyunsaturated fatty acid, g/day	15.3–17.9	16.3–19.9
Carbohydrates, g/day	130.0–150.0	131.0–151.5
Fibre, g/day	20.2–23.2	20.2–23.2
Sodium chloride, g/day	4.0–4.6	4.3–4.9

**Table 2 ijerph-18-07330-t002:** Changes in patients’ anthropometry and blood pressure during the trial, and comparison between groups.

Variables	Control Group							Experimental Group							Between Groups				
	T0	T1	T2		Difference			T0	T1	T2		Difference			T0	T1	T2	Difference	
					T0–T1	T0–T2						T0–T1	T0–T2					T0–T1	T0–T2
	Median (25th–75th) Percentile			P ^a^	Median (25th–75th) Percentile		P ^b^	Median (25th–75th) Percentile			P ^a^	Median (25th–75th) Percentile		P ^b^	P ^c^	P ^c^	P ^c^	P ^c^	P ^c^
Body mass (kg)	103.2 (81.3–128.5)	101.2 (80.1–123.2)	100.3 (79.8–122.1)	0.638	3.1 (1.2–5.4)	2.5 (1.6–6.2)	0.246	99.7 (93.7–123.0)	96.7 (91.7–120.0)	96.5 (92.7–123.7)	0.663	3.7 (2.3–4.5)	3.5 (1.7–5.5)	0.795	0.466	0.420	0.438	0.506	0.752
BMI (kg/m^2)^	39.4 (34.3–43.2)	38.2 (33.4–42.3)	38.7 (33.3–41.7)	0.638	1.2 (0.6–1.8)	0.9 (0.7–2.0)	0.227	41.6 (34.6–47.5)	40.2 (33.0–46.3)	40.0 (31.9–47.4)	0.663	1.4 (0.9–1.8)	1.3 (0.6–2.2)	0.687	0.558	0.580	0.537	0.420	0.837

^a^—based on a Freidman ANOVA, ^b^—based on a two-tailed Wilcoxon test, ^c^—based on a two-tailed Mann–Whitney U test.

**Table 3 ijerph-18-07330-t003:** Changes in patients’ blood pressure during the trial, and comparison between groups.

Variables	Control Group				Experimental Group				Between Groups		
	T0	T2		Difference	T0	T2		Difference	T0	T2	Difference
				T0–T2				T0–T2			T0–T2
	Median (25th–75th) Percentile		P ^b^	Median (25th–75th) Percentile	Median (25th–75th) Percentile		P ^b^	Median (25th–75th) Percentile	P ^c^	P ^c^	P ^c^
SBP (mmHg)	150.0 (140.0–160.0)	130.0 (120.0–135.0)	<0.001	20.0 (15.0–25.0)	150.0 (145.0–165.0)	127.5 (120.0–135.0)	<0.001	25.0 (10.0–30.0)	0.837	0.547	0.189
DBP (mmHg)	90.0 (80.0–95.0)	82.5 (75.0–85.0)	0.001	7.5 (5.0–15.0)	82.5 (80.0–100.0)	75.0 (70.0–80.0)	<0.001	10.0 (5.0–20.0)	0.937	0.133	0.146

^b^—based on a two-tailed Wilcoxon test, ^c^—based on a two-tailed Mann–Whitney U test.

**Table 4 ijerph-18-07330-t004:** Changes in patients’ serum cholesterol profiles during the trial, and comparison between groups.

Variables	Control Group							Experimental Group							Between Groups				
	T0	T1	T2		Difference			T0	T1	T2		Difference			T0	T1	T2	Difference	
					T0–T1	T0–T2						T0–T1	T0–T2					T0–T1	T0–T2
	Median (25th–75th) Percentile			P ^a^	Median (25th–75th) Percentile		P ^b^	Median (25th–75th) Percentile			P ^a^	Median (25th–75th) Percentile		P ^b^	P ^c^	P ^c^	P ^c^	P ^c^	P ^c^
TCL (mg/dL)	142.2 (130.9–162.8)	129.5 (112.6–148.0)	142.7 (135.0–175.9)	0.179	10.9 (−1.5–47.9)	−7.8 (−22.7–13.1)	0.058	158.1 (141.1–214.0)	131.4 (117.4–170.1)	149.6 (124.3–158.4)	0.001	25.0 (14.4–30.5)	31.1 (−2.0–65.6)	0.679	0.056	0.438	0.937	0.261	0.018
CL LDL (mg/dL)	71.8 (60.7–90.1)	66.2 (51.1–83.1)	82.0 (69.8–87.3)	0.029	9.1 (−6.3–29.11)	−8.77 (−18.2–0.7)	0.053	83.7 (73.1–128.4)	62.9 (59.6–102.3)	73.4 (68.8–90.6)	0.003	14.1 (8.1–26.8)	6.6 (0.42–49.1)	0.586	0.074	0.420	0.579	0.248	0.005
CL HDL (mg/dL)	44.2 (39.8–49.3)	41.2 (35.6–47.7)	44.8 (36.9–53.2)	0.029	2.9 (−1.5–6.4)	−2.6 (−7.7–2.6)	0.014	43.0 (35.1–51.6)	38.3 (32.3–44.0)	42.2 (36.1–53.1)	0.017	3.2 (0.0–7.3)	−2.1 (−5.5–2.9)	0.043	0.987	0.351	0.635	0.517	0.537
AI (rel. un.)	2.58 (1.91–3.11)	2.18 (1.75–2.44)	2.44 (1.79–3.08)	0.066	0.31 (−0.21–0.75)	0.12 (−0.48–0.58)	0.078	3.13 (2.53–4.09)	2.59 (1.81–3.42)	2.39 (1.79–2.71)	0.030	0.44 (−0.02–0.86)	0.62 (0.11–1.53)	0.306	0.074	0.079	0.812	0.517	0.064

^a^—based on a Freidman ANOVA, ^b^—based on a two-tailed Wilcoxon test, ^c^—based on a two-tailed Mann–Whitney U test.

**Table 5 ijerph-18-07330-t005:** Changes in patient serum biochemical parameters during the trial, and comparison between groups.

Variables	Control Group							Experimental Group							Between Groups				
	T0	T1	T2		Difference			T0	T1	T2		Difference			T0	T1	T2	Difference	
					T0–T1	T0–T2						T0–T1	T0–T2					T0–T1	T0–T2
	Median (25th–75th) Percentile			P ^a^	Median (25th–75th) Percentile		P ^b^	Median (25th–75th) Percentile			P ^a^	Median (25th–75th) Percentile		P ^b^	P ^c^	P ^c^	P ^c^	P ^c^	P ^c^
Glucose (mg/dL)	101.8 (95.5–118.9)	100.0 (91.9–108.1)	88.2 (82.8–102.7)	0.021	4.5 (−3.6–12.6)	9.0 (1.8–18.0)	0.240	118.9 (100.9–149.5)	112.6 (106.3–131.5)	125.2 (111.7–142.3)	0.084	10.8 (−5.4–12.6)	−9.0 (−14.4–10.8)	0.334	0.057	0.005	<0.001	0.466	0.103
Creatinine (mg/dL)	0.799 (0.738–0.878)	0.770 (0.717–0.937)	0.797 (0.760–0.816)	0.678	−0.016 (−0.137–0.086)	0.008 (−0.072–0.094))	0.557	0.850 (0.767–0.952)	0.883 (0.791–1.078)	0.852 (0.772–0.912)	0.128	−0.052 (−0.180–0.027)	−0.018 (−0.053–0.085)	0.028	0.275	0.048	0.071	0.384	0.987
Urea (mg/dL)	39.1 (33.5–51.8)	38.9 (32.8–47.2)	42.5 (34.9–49.0)	0.167	3.8 (−5.5–8.5)	−2.8 (−9.6–2.5)	0.227	46.1 (35.3–51.6)	53.5 (37.4–85.2)	46.9 (41.9–59.3)	0.042	−10.9 (−32.4– –1.6)	−5.8 (−16.0–8.5)	0.035	0.477	0.008	0.150	0.003	0.764
LDH (U/L)	136.8 (107.8–160.0)	134.0 (104.1–150.8)	143.7 (125.6–153.9)	0.358	7.2 (−19.3–32.9)	−3.9 (−18.6–6.1)	0.071	131.3 (116.1–157.8)	124.9 (98.7–137.9)	138.6 (124.1–152.3)	0.115	11.8 (−1.4–32.9)	−11.9 (−25.1–18.9)	0.025	0.812	0.578	0.800	0.411	0.888
CPK (U/L)	77.7 (66.7–106.4)	78.4 (59.2–93.1)	75.3 (61.8–97.9)	0.801	−1.2 (−38.4–23.1)	1.9 (−4.8–43.2)	0.679	80.9 (65.4–96.3)	89.2 (55.2–119.0)	85.3 (56.0–113.6)	0.678	−4.0 (−51.2–6.3)	1.2 (−27.6–32.1)	0.396	0.912	0.602	0.402	0.384	0.477
AlkPhos (U/L)	99.3 (83.8–108.5)	95.6 (86.6–103.0)	96.5 (94.6–98.8)	0.678	2.0 (−2.2–9.2)	1.8 (−9.1–10.0)	0.349	80.7 (71.0–96.9)	90.2 (83.8–101.6)	86.0 (79.0–105.1)	0.066	−4.1 (−13.8–0.7)	−4.7 (−30.4–0.2)	0.500	0.094	0.558	0.141	0.031	0.052
GGTP (U/L)	21.7 (13.4–38.0)	18.2 (13.2–33.1)	26.0 (15.6–60.3)	0.486	1.8 (−1.3–5.3)	−1.5 (−7.8–3.1)	0.231	20.6 (16.4–30.8)	19.6 (15.7–26.6)	24.1 (16.6–31.4)	0.211	2.0 (−0.3–4.2)	0.0 (−2.7–2.9)	0.151	0.888	0.728	0.517	0.558	0.692
AST (U/L)	20.6 (15.6–24.1)	21.0 (16.0–24.0)	18.2 (16.2–21.7)	0.946	1.1 (−2.1–3.6)	−0.6 (−2.7–3.2)	0.862	18.5 (15.1–24.0)	19.7 (15.8–24.7)	19.2 (15.4–22.5)	0.577	−0.4 (−2.2–1.5)	−0.2 (−4.0–2.5)	0.500	0.624	0.862	0.602	0.467	0.924
ALT (U/L)	17.7 (10.2–23.7)	15.4 (11.2–32.0)	20.4 (13.2–29.8)	0.607	–0.9 (−6.6–5.0)	−2.2 (−11.2–5.7)	0.647	13.6 (10.5–16.9)	13.6 (9.7–19.8)	12.1 (9.3–20.1)	0.678	−1.6 (−5.5–4.8)	−2.0 (−5.5–2.9)	0.913	0.235	0.319	0.154	0.937	0.987
De Ritis ratio (rel. un.)	1.24 (1.01–1.46)	1.28 (0.72–1.70)	0.96 (0.74–1.31)	0.311	0,10 (−0.46–0.46)	0.26 (−0.46–0.48)	0.184	1.34 (1.08–2.13)	1.42 (0.85–2.37)	1.43 (0.99–1.61)	0.846	−0.09 −0.79–0.73)	0.22 (−0.35–0.74)	0.616	0.261	0.223	0.133	0.937	0.837

^a^—based on a Freidman ANOVA, ^b^—based on a two-tailed Wilcoxon test, ^c^—based on a two-tailed Mann–Whitney U test.

**Table 6 ijerph-18-07330-t006:** Correlation between patient serum cholesterol profile and anthropometry and blood pressure during the trial.

Variables	Control Group				Experimental Group			
	Cholesterol, mg/dL			AI (rel. un.)	Cholesterol, mg/dL			AI (rel. un.)
	Total	LDL	HDL		Total	LDL	HDL	
	Difference T0–T2 (outpatient Period)							
Body mass (kg)								
Correlation coefficient	0.261	0.355	−0.126	0.436	0.411	0.404	0.553 *	0.069
Significance (two-tailed)	0.295	0.148	0.618	0.071	0.090	0.097	0.017	0.785
BMI (kg/m^2^)								
Correlation coefficient	0.304	0.395	−0.067	0.432	0.422	0.391	0.534 *	0.086
Significance (two-tailed)	0.219	0.104	0.791	0.073	0.081	0.109	0.024	0.735
SBP (mmHg)								
Correlation coefficient	0.093	0.288	−0.440	0.240	−0.050	−0.069	0.075	−0.218
Significance (two-tailed)	0.713	0.246	0.067	0.336	0.842	0.784	0.768	0.386
DBP (mmHg)								
Correlation coefficient	−0.012	0.156	0.130	−0.164	−0.017	0.000	0.184	−0.243
Significance (two-tailed)	0.963	0.537	0.606	0.515	0.947	1.000	0.465	0.331

* Statistically significant.

**Table 7 ijerph-18-07330-t007:** Correlation between patients’ serum atherogenic index and cholesterol profiles during the trial.

Variables	Control Group			Experimental Group		
	Cholesterol, mg/dL					
	Total	LDL	HDL	Total	LDL	HDL
	Difference T0–T1 (inpatient period)					
AI (rel. un.)						
Correlation coefficient	0.525 *	0.476 *	0.003	0.600 *	0.507 *	−0.238
Significance (two–tailed)	0.025	0.046	0.990	0.009	0.032	0.341
	Difference T0–T2 (outpatient period)					
AI (rel. un.)						
Correlation coefficient	0.670 *	0.703 *	0.020	0.740 *	0.810 *	0.071
Significance (two–tailed)	0.002	0.001	0.938	0.001	<0.001	0.779

* Statistically significant.

**Table 8 ijerph-18-07330-t008:** Correlation between patients’ age, gender and cholesterol profiles during the trial.

Variables	Control Group			Experimental Group		
	Cholesterol, mg/dL					
	Total	LDL	HDL	Total	LDL	HDL
	Difference T0–T1 (inpatient period)					
Age						
Correlation coefficient	−0.149	−0.158	0.070	−0.163	−0.181	−0.332
Significance (two-tailed)	0.556	0.531	0.784	0.518	0.472	0.178
Gender						
Correlation coefficient	0.209	0.165	0.231	−0.209	−0.318	−0.055
Significance (two-tailed)	0.406	0.513	0.357	0.406	0.198	0.829
	Difference T0–T2 (outpatient period)					
Age						
Correlation coefficient	−0.213	−0.194	−0.216	–0.119	−0.175	−0.137
Significance (two-tailed)	0.396	0.440	0.390	0.638	0.488	0.587
Gender						
Correlation coefficient	−0.055	−0.165	0.022	−0.209	−0.253	−0.286
Significance (two-tailed)	0.829	0.514	0.931	0.406	0.312	0.250

## Data Availability

The datasets generating and analysed during the current study are not publicly available, due to ethical reasons, but are available from the corresponding author upon a reasonable request.

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
