# Peer review of "Bioactive Compounds of Porcine Hearts and Aortas May Improve Cardiovascular Disorders in Humans"

_ijerph, 2021, doi:10.3390/ijerph18147330_

Round 1
Reviewer 1 Report
This study examined the nutritional make-up of a specialized meat product and the impact of consuming this meat product with a low-calorie randomized-controlled trial. Overall, the study examines a novel food in a reasonable way. My comments are primarily about missing content that would should be included such as limitations, overall design of the study and the more concise conclusions.Introduction:
-The study aim should address all the aims of the study- including examining the nutritional content of the meat product. Methods: -Some clarity in the RCT design would be useful. How were the subjects consuming the outpatient diet? Were the counseled to consume certain foods or was the outpatient food provided daily to the participants? If they were following the outpatient diet themselves, then a discussion on compliance would be important. Additionally, as inclusion was based on being hospitalized for diet treatment, further explanation on this would be relevant. Also, the inclusion criteria is not descriptive enough- how was atherosclerosis defined? Results: -Differences between groups: the only differences between groups examined were anthropometric. However, the authors do not discuss whether there were differences in other relevant confounders of metabolic health including age, sex, or race/ethnicity. This is an important point of discussion as these are all relevant confounders which may have influenced the results. Discussion: -It would be useful and more concise if the first paragraph of the discussion framed the overall results before referencing results from other studies. -The authors have not listed any limitations to this research, which is an important element of all studies. Some limitations worth noting: confounding by sex, race/ethnicity, socioeconomic status, physical activity level, weight history, income, etc. -Table 6 and Table 7 belong in the results. This is still results and the content referencing them should also be in the results. Minor comments: -line 39-41: this needs a reference and could be re-worded for clarity (line 40-41) -line 64-65: This could also be re-worded for clarity, it is unlikely any food would “relieve” metabolic syndrome. Do you mean to say “prevent”? It is also common practice now to use “people-first” language in terms of obesity, instead of “obese people”, “people with obesity”.Author Response
Dear Reviewer 1,
On behalf of the co-authors I thank the Reviewer 1 for the valuable comments. Please see the attachment.
Best regards,
Authors.

Reviewer 2 Report
Interesting and well-founded theme.
In the methods, point 2.5 blood sampling and following needs to clarify the concept and type of collection. There seems to be an error that remains throughout the article regarding the biological fluid used which is described as serum, but if the collection is done into an EDTA tube what you get is plasma.
The EDTA used (K3EDTA or Na2EDTA) should also be more specific. So review and if EDTA was used then in line 135-replace coagulant by anticoagulant.
line 139 withdraw serum
line 146 if you collect with EDTA what you get is plasma
the word serum should be revised in lines 224, 241, 244, 250, 254, 262, 299, 310, 408, 410, 412, 420, 481 through 500.
Tables 2, 3, 4. Presentation of results is confusing. Reducing the font size I think improves the visualization and understanding of the results.
Author Response
Dear Reviewer 2,
On behalf of the co-authors I thank the Reviewer 2 for the valuable comments. Please see the attachment.
Best regards,
Authors.
